# The COVID-19 Pandemic and Responses in Nursing Homes: A Cross-Sectional Study in Four European Countries

**DOI:** 10.3390/ijerph192215290

**Published:** 2022-11-19

**Authors:** Fabrice Mbalayen, Valentine Dutheillet-de-Lamothe, Aude Letty, Solenn Le Bruchec, Manon Pondjikli, Gilles Berrut, Lamia Benatia, Biné Mariam Ndiongue, Marie-Anne Fourrier, Didier Armaingaud, Loic Josseran, Elisabeth Delarocque-Astagneau, Sylvain Gautier

**Affiliations:** 1UFR Simone Veil–Santé, Université Versailles Saint-Quentin-en-Yvelines, 78180 Montigny-le-Bretonneux, France; 2Département Hospitalier d’Épidémiologie et de Santé Publique, Hôpital Raymond-Poincaré, Groupe Hospitalier Universitaire Université Paris-Saclay, Assistance Publique-Hôpitaux de Paris, 92380 Garches, France; 3Gérontopôle Autonomie et Longévité des Pays de la Loire, 44000 Nantes, France; 4Fondation Korian pour le Bien-Vieillir, 75000 Paris, France; 5Pôle Hospitalo-Universitaire de Gérontologie Clinique, Centre Hospitalier Universitaire de Nantes, 44000 Nantes, France; 6Direction Médical Éthique et Qualité, Groupe Korian, 75000 Paris, France; 7Centre de Recherche en Épidémiologie et Santé des Populations, UMR 1018, Université de Versailles-Saint-Quentin-en-Yvelines, Université Paris-Saclay, 75180 Montigny-Le-Bretonneux, France

**Keywords:** COVID-19, mortality, nursing home, elderly, first wave

## Abstract

Background: Studies comparing how the European nursing homes (NHs) handled the first wave of the COVID-19 pandemic remain scarce. Methods: A cross-sectional study was conducted during the first wave in a private NHs network in Belgium, France, Germany and Italy. Mortality rates were estimated, and prevention and control measures were described by country. Data from the Oxford governmental response tracker project were used to elaborate a “modified stringency index” measuring the magnitude of the COVID-19 global response. Results: Of the 580 NHs surveyed, 383 responded to the online questionnaire. The COVID-19 mortality rate was similar in France (3.9 deaths per 100 residents) and Belgium (4.5). It was almost four times higher in Italy (11.9) and particularly low in Germany (0.3). Prevention and control measures were diversely implemented: residents’ sectorization was mainly carried out in France and Italy (~90% versus ~30% in Germany and Belgium). The “modified stringency index” followed roughly the same pattern in each country. Conclusion: This study, conducted in a European network of NHs, showed differences in mortality rate which could be explained by the characteristics of the residents, the magnitude of the first wave and the prevention and control measures implemented. These results may inform future European preparedness plans.

## 1. Introduction

The first wave of the COVID-19 epidemic reached western European countries at the beginning of the year 2020. Italy, France and Belgium were quickly and strongly hit, while other countries such as Germany were less affected. However, their overall first wave epidemic profiles are comparable [1], due to a similar demography of the aging population. This is all the more important since elderly people represented the main at-risk population for developing severe COVID-19. Transmission risk was also deemed higher among elderly people living in nursing homes [2]. Estimates suggest that nursing home (NH) residents may account for 50% or more of COVID-19 deaths in Europe. However, there were some disparities between western European countries: from 38% in Germany to 66% in Spain and 50% in Belgium and in France [3].

These disparities may reflect differences in NH characteristics and residents’ profiles, as some studies provided evidence on their respective role in the burden of COVID-19 death in NH [4,5]. In addition, general prevention and control measures were the only way to mitigate the COVID-19 impact during the first wave, in the absence of a cure or a vaccine. These measures were issued both at a national level through the different policies decided by the public health authorities and implemented at the NH level [6]. In addition, previous viral pandemics (e.g., H1N1) led to national preparedness plans upon which the NHs also built their own response to the epidemic. They also benefited from their experience gained during previous seasonal viral epidemics. Thus, measures taken in NHs may have slightly differed between countries.

Studies comparing how the NHs handled all the dimensions of COVID-19 epidemic in their premises, between European countries, remain scarce. In this context, the European Korian network of private NHs represented an opportunity to investigate the burden of the pandemic and the measures implemented, in a comparative way. Thus, the aim of this study was to describe, on the one hand, the impact of the COVID-19 pandemic in NHs in terms of mortality and hospitalization and, on the other hand, the responses in a network of NHs of four European countries.

## 2. Methods

This observational cross-sectional study was conducted during the first wave of the epidemic in a European private NHs network in four countries: Belgium, France, Germany and Italy, composed of 73 NHs, 290 NHs, 177 NHs and 40 NHs, respectively, corresponding to a total of 51,356 residents.

### 2.1. Data Collected

Data were collected between 14 September 2020 and 27 October 2020 from an online questionnaire covering the period from 1 February 2020 to 31 July 2020. It was sent to each NH of the network in the above-mentioned countries and translated from French to the corresponding language. Information collected focused on the main organizational and structural characteristics of each NH (number of beds, surface, presence of specific units such as “Alzheimer unit” …), on the characteristics of the residents (aggregated data on mean age, dependency level…) and details on the management of the COVID-19 epidemic (hospitalizations requests, prevention and control measures taken…) as well as COVID-19 death numbers. The dependency level of the residents was assessed using a different scale in each country: the Katz scale in Belgium [7], the GIR scale in France [8], the “Begutachtungs-instrument” in Germany [9] and the Barthel index in Italy [10]. COVID-19 deaths were defined as deaths occurring during the first epidemic wave, confirmed by a PCR-test or suspected by a physician. Among the prevention and control measures taken by the NHs, information on visitor bans, residents’ containment, sectorization, implementation of dedicated COVID-19 units, staff and residents’ screening, training in hygiene practices were collected. The different definitions of the prevention and control measures implemented are detailed in Appendix A.

### 2.2. Other Sources of Data

In order to contextualize potential differences between the countries studied, we used data from the Oxford governmental response tracker project [11] lead by Thomas Hale et al. In this project, the researchers computed 23 indicators reflecting the daily level of the different policy responses during the COVID-19 pandemic in 180 countries. From these indicators, they developed several indices, such as the “stringency index”, which measures the level of the response at a given time in a given country. This original “stringency index” is calculated from the nine following response indicators: (1) school closures; (2) workplace closures; (3) cancellation of public events; (4) restrictions on public gatherings; (5) closures of public transport; (6) stay-at-home requirements; (7) public information campaigns and (8) restrictions on internal movements and (9) international travel controls. The “stringency index”, on any given day, is calculated as the mean score of these nine indicators, each taking a value between 0 and 100, a higher score indicating a stricter response.

Here, we computed a “modified stringency index”. In order to consider an index more suited to the context of our study, we added the specific “protection of elderly people” indicator (H8) and the “testing policy” indicator (H2). Moreover, we did not include the “stay at home requirements” indicator (C6) and the “public information campaign” indicator (H1) as we considered them not sufficiently discriminant between the countries studied. The calculation of this “modified stringency index” was based on the same calculation process developed by the Oxford team [11]. Finally, we obtained a daily quantitative bounded variable between 0 to 100 for Belgium, France, Germany and Italy. This index recorded the strictness of government policies over the period from 1 February to 31 July 2020. The time period considered corresponded to the period in which we collected the mortality data in the participating NHs. We presented weekly data using the last available value of a given week.

### 2.3. Statistical Analysis

The statistical unit of the study was the NH. Descriptive statistics were performed to quantify the implementation of the different prevention and control measures for the epidemic, as well as mortality and hospitalization related to COVID-19 within the “NHs” for each country of the study. Numbers and proportions were used to describe qualitative variables and sum or mean (with standard deviation) for quantitative variables. Categorical and quantitative variables were compared using chi squared tests and one-way ANOVA, respectively. A *p* value of less than 0.05 was regarded as statistically significant. The COVID-19 mortality rate and 95% confidence interval were estimated for each country as the number of reported COVID-19 deaths over the number of residents. All statistics were conducted using the free software *R*, version 3.6.2.

## 3. Results

Of the 580 NHs of the network surveyed, 383 (66.0%) responded to the online questionnaire. Participation was 100%, 61.7%, 66.2% and 48.0% in Italy, Belgium, France and Germany, respectively. Resident mean age was higher in France (88 ± 1.8), lower in Germany (80 ± 6.1) and quite similar in Belgium and Italy (85 ± 3.9 and 84 ± 4.0, respectively) (Table 1). Resident mean age was significantly different between countries (*p* < 0.01). The proportion of dependent residents was higher in Germany (80.0%) and Italy (70.1%) and lower in Belgium (49.4%) and France (53.7%) (*p* < 0.01).

The proportion of NHs reporting a COVID-19 cluster close to their premises varied from 65.0% in Italy to 24.2% in Belgium. The proportion of accepted hospitalization requests among NHs who reported at least one COVID-19 case in France was lower (68.5%) than in Germany (100.0%) and Belgium (84.4%) (*p* < 0.01). The COVID-19 mortality rate as number of COVID-19 deaths per 100 residents was similar in France (3.9, 95% CI = (2.9–4.9)) and Belgium (4.5, 95% CI = (2.9–6.2)), whereas it was almost four times higher in Italy (11.9, 95% CI = (8.1–15.9)). In Germany, the mortality rate was particularly low (0.3, 95% CI = (0.05–0.6)).

In almost all the NHs of the four countries, visitors were banned from accessing the nursing home premises (Table 2). Residents were confined in their room in 100%, 99.4%, 18.1% and 57.5% of the nursing homes in Belgium, France, Germany and Italy, respectively. Residents’ sectorization was implemented in circa one-third of nursing homes in Belgium and Germany and reached 90% or more in France and Italy. In addition, implementation of COVID-19 units in the nursing homes was heterogeneous between the four countries and did not always include a dedicated day/night staff. In each country, more than 90% of the facilities had conducted standard precautions’ training in the previous 2 years. An external hygiene team was in support in at least half of the NHs. In the same way, more than 90% (except France at 73.4%) of the NHs reported that an audit of practices had been carried out. Systematic screenings were performed among residents and staff in circa 90% in Belgium, France and Italy. In Germany, it was carried out in less than one-third of the NHs.

### Containment and Closures Policies Indicators

Overall, the start of implementation of these policies varied between countries, with Italy always starting sooner (Figure 1). For instance, “workplace closing” took place at week 7 of the year 2020 in Italy whereas it took place 3 weeks later for Belgium (week 10) and 4 weeks later for France and Germany (week 11). In the same way, Italy started with a high level of “protection of elderly people” (“extensive restrictions for isolation and hygiene in long term care facilities, all non-essential external visitors prohibited”) followed by Germany 1 week after with an increase in two phases, by France 2 weeks after, and then, by Belgium 3 weeks after. In addition, the level of the measures varied: regarding “school closure” indicator, France and Italy implemented the measure at the highest level (“require closing all school degrees”) whereas Germany and Belgium started at the medium level (“only some school degrees”). Regarding the “protection of elderly people”, the duration differed between countries: the measure remained in place at a medium level (“narrow restrictions for isolation, hygiene in long term care facilities”) during more than 10 weeks with variations between countries. Resulting from the combination of these indicators, the “modified stringency index” followed roughly a same pattern, with a prompt increase to the highest index value (between 70 and 80%) at different weeks and then a stabilization around an index value estimated at 60% for the four countries until 31 July 2020.

## 4. Discussion

This study, conducted in a large private European network of NHs, showed differences in mortality rate across the participating countries, with a higher mortality in Italy, comparable figures in France and Belgium and a much lower mortality in Germany. Residents’ characteristics also differed: in French NHs, residents were older and less dependent, when in German NHs, they were younger and more dependent. In addition, prevention and control measures were largely reported, with disparities between countries on their completeness of coverage.

The COVID-19 mortality in NHs is expected to be in relation with the magnitude of the epidemic, the timeliness and stringency of the prevention and control measures issued at the national level and, at a local level, the measures implemented in the NH and the characteristics of the premises and the residents. In particular, there are several hypotheses to explain the higher COVID-19 mortality rate observed in the French NHs surveyed compared to German ones. First, strategies in detecting COVID-19 cases differed, with a more systematic approach in France than in Germany. Indeed, in the latter, it was recommended to test residents and staff only in the occurrence of a suspected case. Second, the definition of a COVID-19 death varied between the two countries, with a more sensitive definition used in French NHs [12]. Third, the magnitude of the epidemic was higher in France, thus increasing the risk of introduction of the virus in NH [13,14]. In the NHs, face masks were available later in France. A contrario, Germany had built up a stockpile of protective masks and was implementing a policy of banning the export of its protective equipment [15]. Finally, mean age of residents in the German NHs was lower, with a difference of about 8 years, thus decreasing the risk of death. In addition, the proportion of accepted hospitalization requests may also have affected the mortality in NHs: this proportion was lower in France than in Germany. However, the number of requests in France largely exceeded the number of requests in Germany. This raises the question of the capacity of the NHs to take care of COVID-19 patients, which could differ between the countries. Indeed, in Germany, it has been shown that the majority of NH managers reported no deficits in general practitioners’ [16].

In this study, details on the measures put in place in the NHs were collected. First of all, in order to limit the risk of introduction of the virus in the premises, visits were banned in almost all the surveyed NHs from the four countries. To decrease the risk of transmission inside the NH, residents’ containment measure was widely implemented, whether or not there was already a case of COVID-19 detected in the premises. However, in Germany and Italy, this measure was less reported. In Italy, where the majority of residents were hosted in double rooms, the containment measure could not be possible. Thus, sectorization was implemented. In Germany, this measure was poorly reported, partly because of a lower incidence of COVID-19 in this country [14]. In the NHs where at least one COVID-19 case occurred, NHs set up a COVID-19 unit with dedicated staff. Regarding hygiene practices, standard precautions’ training and audit of practices were widely carried out. Nevertheless, the support by an external hygiene team was less reported, while it allows the advice issued to be updated in a rapidly changing context.

All these measures of prevention and control of the epidemic must be put in parallel with the temporality of the series of measures issued at the national level of each country, in relation with the specific epidemic context. All the data collected by the Oxford governmental response tracker project make it possible to appreciate the timing and the magnitude of the policy responses (such as “school closure”, “restriction on gatherings”, “cancel public events”, “restriction on movement”, “protection of elderly people” …) and to compare them between the countries considered. The “protection of elderly people” indicator completes the information on the types of specific measures implemented in the NHs. Indeed, it shows that the timing was slightly different between the countries as well as the duration of this measure. In addition, the “modified stringency index”, calculated as a combination of these indicators, provides an overview of the measures. Despite an earlier and a more stringent response observed in Italy, the mortality in the surveyed NHs was the highest. This likely reflects the fact that Italy was the first European country affected by the epidemic.

This study has several limits. Due to its cross-sectional design, we only provided cumulative mortality estimates and the timing of the prevention and control measures was not available, which makes it not possible to establish a relation between the mortality and the measures. Information bias is possible due to the retrospective nature of the data collection. In addition, different dependency scales were used. The study only concerned a network of private NHs which are not representative of all the NHs in the countries considered. In addition, although all the NHs belonging to the network in Italy participated in the study, the number is limited. However, the COVID-19 mortality estimates and the magnitude of the difference between France and Germany are comparable to the figures published by the ECDC Public Health Emergency Team [3].

## 5. Conclusions

Despite these limitations, this study offers a descriptive approach of the impact and management of the COVID-19 epidemic in NHs in four European countries, mobilizing a “modified stringency index” measuring the magnitude COVID-19 global response at the country level. It enriches the lessons-learned process at the European scale. More evidence is, however, needed to improve the next preparedness plans on emerging diseases dedicated to the NHs, in a European perspective. Some differences between the NHs surveyed, such as the characteristics of the residents (age, dependency…) and the measures taken, underline the necessity to adapt the plans at the local level. In addition, as a perspective, the stringency of the measures implemented in the NHs should take into account the magnitude of the local epidemic.

## Figures and Tables

**Figure 1 ijerph-19-15290-f001:**
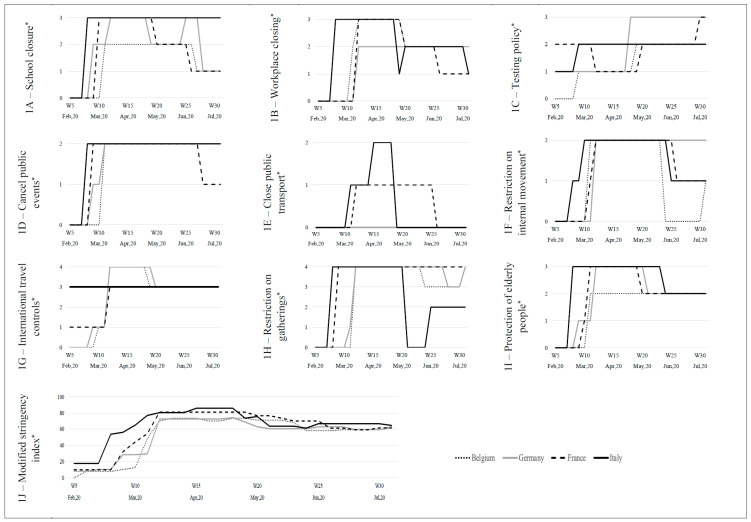
Containment and closure policies indicators and “Modified stringency index” in Belgium, France, Germany, Italy: trends between 1 February and 31 July 2020. * **School closure**: (0) none; (1) recommend closing; (2) require closing only some levels or categories; (3) require closing all levels. **Workplace closing**: (0) none; (1) recommend closing; (2) require closing only some sectors or categories of workers; (3) require closing for all but essential. **Testing policy**: (0) none; (1) only those who both have symptoms meet specific criteria; (2) testing of anyone showing COVID-19 symptoms; (3) open public testing. **Cancel public events**: (0) none; (1) recommend cancelling; (2) require cancelling. **Close public transport**: (0) none; (1) recommend closing; (2) require closing. **Restrictions on internal movement**: (0) none; (1) recommend not to travel between regions or cities; (2) internal movement restrictions in place. **International travel controls**: (0) none; (1) screening arrivals; (2) quarantine arrivals from some oral regions; (3) ban arrivals from some regions; (4) ban on all regions or total border closure. **Restrictions on gatherings**: (0) none; (1) restrictions on very large gatherings (the limit is above 1000 people); (2) restrictions on gatherings between 101 and 1000 people; (3) restrictions on gatherings between 11 and 100 people; (4) restrictions on gatherings of 10 people or less. **Protection of elderly**: (0) none; (1) recommended isolation, hygiene and visitor restriction measures in long term care facilities and/or elderly people to stay at home; (2) narrow restrictions for isolation, hygiene in long term care facilities; (3) extensive restrictions for isolation and hygiene in long term care facilities, all non-essential external visitors prohibited. **Modified stringency index**: composite measure based on nine response indicators rescaled to a value from 0 to 100 (100 = strictest). This index records the strictness of government policies over the period from 1 February to 31 July 2020.

**Table 1 ijerph-19-15290-t001:** Main characteristics of nursing homes (NHs) surveyed, mortality rates and hospitalization for COVID-19. CluDe Study, 2020.

	Belgium N = 66	France N = 192	Germany N = 85	Italy N = 40
**Total number of residents**	6422	15,307	8399	4412
*Dependent residents ^1^, n (%)*	*3173 (49.4)*	*8227 (53.7)*	*6723 (80.0)*	*3091 (70.1)*
*Partially dependent residents, n (%)*	*2559 (39.8)*	*5679 (37.1)*	*1658 (19.7)*	*841 (19.1)*
*Autonomous residents, n (%)*	*690 (10.7)*	*841 (5.5)*	*50 (0.6)*	*480 (10.9)*
**Residents’ mean age, mean ± SD**	85 ± 3.9	88 ± 1.8	80 ± 6.1	84 ± 4.0
**Reporting a COVID-19 cluster close to the NH, n (%)**	16 (24.2)	75 (39.0)	27 (33.8) ^2^	26 (65.0)
**Number of NHs with at least one COVID-19 case**	44 (66.7)	125 (65.1)	15 (17.6)	22 (55.0)
**Number of hospitalization requests for COVID-19 cases** **Accepted hospitalization requests, n (%)**	224 189 (84.4)	603 416 (68.5)	30 30 (100.0)	N.A. ^3^ N.A.
**COVID-19 mortality rate: number of deaths for 100 residents, mean [95% CI] ^4^**	4.5 (2.9–6.2)	3.9 (2.9–4.9)	0.3 (0.05–0.6)	11.9 (8.1–15.9)

**^1^** The dependency level of the residents was assessed using the Katz scale in Belgium, the GIR scale in France, the “Begutachtungs-instrument” in Germany and the Barthel index in Italy. ^2^ Reporting a COVID-19 cluster close to the nursing home: 5, missing data for Germany. ^3^ N.A.: not available. ^4^ (95% CI): 95% confidence interval.

**Table 2 ijerph-19-15290-t002:** Characteristics of prevention and control measures implemented in the nursing homes surveyed. CluDe Study, 2020.

*n* (%)	Belgium N = 66	France N = 192	Germany N = 85	Italy N = 40
**Visitor ban**	66 (100.0)	192 (100.0)	81 (95.3)	40 (100.0)
**Residents’ containment ^1^**	66 (100.0)	189 (99.4) ^2^	15 (18.1) ^2^	23 (57.5)
**Residents’ sectorization ^1^**	25 (37.9)	177 (92.1)	29 (34.5) ^3^	38 (95.0)
**Dedicated COVID-19 unit ^1^**	29 (70.7)	104 (92.0)	9 (60.0)	19 (90.5)
***Day staff dedicated to this unit***	*25*	*98*	*8*	*17*
***Night staff dedicated to this unit***	*13*	*80*	*8*	*10*
**Support by an external hygiene team**	62 (93.9)	123 (64.0)	43 (50.6)	19 (47.5)
**Standard precautions’ training in the previous 2 years**	62 (93.9)	187 (97.3)	80 (94.1)	39 (97.5)
**Audit of practices**	62 (93.9)	141 (73.4)	79 (94.0) ^4^	40 (100.0)
**Residents’ screening ^5^**	62 (93.9)	179 (93.2)	18 (22.5) ^6^	34 (85.0)
**Staff’ screening**	63 (95.5)	184 (95.8)	24 (29.3) ^7^	33 (82.5)

^1^ Residents’ containment was defined as confining the residents to their room. Residents’ sectorization was defined as partitioning areas/spaces of the nursing home premises with clear restrictions on who can access it. A dedicated COVID-19 unit is an area where COVID-19 positive tested patients are taken care of. COVID-19 unit was implemented when there was at least one confirmed case of COVID-19 in the nursing home. The number of COVID-19 units was only reported on NHs where at least 1 case of COVID-19 occurred: 41, 113, 15 and 21 nursing homes reported at least 1 case of COVID-19 in Belgium, France, Germany and Italy, respectively. ^2^ Residents’ containment variable: 2 missing data points for France and Germany, respectively. ^3^ Residents’ sectorization variable: 1 missing data point for Germany. ^4^ Audit of practices: 1 missing data point for Germany. ^5^ In Germany, a screening was only carried out by the NH among staff members and residents if there were suspicious cases. It concerned 125 NHs. ^6^ Residents’ screening: 5 missing data points for Germany. ^7^ Staff screening: 3 missing data points for Germany.

## Data Availability

Data were provided by the Korian Foundation for Ageing Well.

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
