# Peer review of "The COVID-19 Pandemic and Responses in Nursing Homes: A Cross-Sectional Study in Four European Countries"

_ijerph, 2022, doi:10.3390/ijerph192215290_

Round 1
Reviewer 1 Report
It was a pleasure to review the study. I think this study provides a good summary of experience on how nursing homes can prevent and control COVID-19. It is suggested to beautify and adjust the title and label of Figure 1.
Reviewer 2 Report
General
The article addresses an important and relevant topic of understanding the COVID-19 pandemic responses in nursing homes (NHs). Indeed, there were differences among countries in facing the pandemic and it is important to evaluate those differences. The present article describes the differences among countries although unfortunately there are no results on the potential relationships between different responses and relevant outcomes due to lack of appropriate data.
- I believe describing the pandemic responses across Europe is interesting and important in itself, even if it is not always possible to correlate it with the outcomes.
- I would suggest perhaps considering changing a title since it now implies that the article is about the relationship between NH responses and impact (outcomes) of pandemic?
Detailed
Abstract
Line 18: I suggest the NHs abbervation should be written down when first used in the abstract.
Introduction
Line 57: I suggest rephrasing the aim of the study to be more clear, what is compared in the article. I believe there is comparison between countries but there is no comparison of responses to the outcomes? (I acknowledge that that is already mentioned in the limitations part.)
Methods
Line 81: Among the prevention and control measures taken by the NHs, information on visitor bans, residents’ containment, sectorization, implementation of dedicated COVID-19 units, staff and residents’ screening, training in hygiene practices…
I think there is a verb missing in that sentence perhaps?
Line 92: Modified stringency index should be further explained. It is explained how it was modified but there is nothing about what original stringency index is? There is a source for calculation, nevertheless I believe some short explanation is needed, perhaps at least the domains that are included in the calculation? Perhaps it should also be added what stringency index measures? Also if this stringency index was validated before as a marker of epidemic (response) than this would add an important aspect to this article.
Results
Line 120: Table 1. I suggest more careful use of the abbreviations NH and NHs throughout the article. For example in Table 1 it should probably be “Number of NHs with at least one COVID-19 case”.
Line 145: Table 2. At first I couldn’t find the explanation for superscript numbers in Table 2. I found them at the end of the article but that is probably just a formatting mistake?
Line 148: Is that a sentence in the results or a subtitle or something related to the table 2? I believe it is unclear at the moment.
Line 150: Would it be possible to present the data as a relative time from the start of pandemic in a specific country? Currently the time scale is the same for all countries but the state of epidemic in each country was probably different with Italy being the first outbreak in Europe. Perhaps the timing of the responses in NHs relative to the start of epidemic in a specific country would better correlate with the outcomes (mortality)?
Figure 1. Currently the Figure is on two pages with only one legend. If there are two pages for the Figure, I suggest putting a legend on each page or putting the entire figure on 1 page. Also there is a missing y axis title for the first chart? And the meaning of 1G, 1D should be explained somewhere or omitted.
Discussion
Line 177: The first sentence of this second paragraph – do the results of the study explore the relationship between mortality and NHs responses or characteristics of residents? If yes, than this should be more clearly presented in the results part. If not, than this sentence/paragraph in the discussion should be supported with external sources or rephrased perhaps?
Line 205: You mention here a lower incidence of COVID-19 in Germany – is that perhaps the main reason for low mortality in Germany? Together with timely responses? Would it be possible or feasible to include the state of COVID-19 epidemic in each country at the time of the collection of other data in this article?
Line 221: Despite an earlier and a more stringent response observed in Italy, the mortality in the surveyed NHs was the highest. This likely reflects the fact that Italy was the first European country affected by the epidemic.
I believe this is probably the right observation. And perhaps this is why presenting the time relative to the start of pandemic in each country or perhaps just including the data on the state of pandemic in a specific country (or region of a country with included NHs?) could benefit the results. In the conclusion there is a sentence: “In addition, the stringency of the measures implemented in the NHs should take into account the magnitude of the local epidemic”. I agree with that observation and suggest including those data in the article if feasible or otherwise simply presenting and discussing the NH responses in different countries, which is also an important observation in itself (without discussing correlations to the outcomes, if not supported by the results).
Line 224: I suggest mentioning in the limitations part, that different dependency scales were used in different countries.
Reviewer 3 Report
The study and its findings are interesting and need of the hour.
however, the quality of the paper can be further enhanced by the following.
1) Please indicate on sampling procedure and sample size determination
2) The data collected from Italy is small, hence please elaborate in the discussion and results when reporting in 100% which may mislead the readers.
3) the statistical analysis can be improved, its correlation with the sociodemographic data can be included to make the findings more impactful.
4) authors can try to check the association of mortality with the preventive measures/strategies
all the best
Round 2
Reviewer 2 Report
I have no further comments.
Author Response
Thank you very much for your feedback.
Reviewer 3 Report
Authors were unable to address all the comments. specially the statistical test requested which are basics for any good paper.
